# SiC@FeZnZiF as a Bifunctional Catalyst with Catalytic Activating PMS and Photoreducing Carbon Dioxide

**DOI:** 10.3390/nano13101664

**Published:** 2023-05-18

**Authors:** Zhiqi Zhu, Liaoliao Yang, Zhaodong Xiong, Daohan Liu, Binbin Hu, Nannan Wang, Oluwafunmilola Ola, Yanqiu Zhu

**Affiliations:** 1Key Laboratory of Disaster Prevention and Structural Safety of Ministry of Education, Guangxi Key Laboratory of Disaster Prevention and Engineering Safety, College of Chemistry and Chemical Engineering, School of Resources, Environment and Materials, Guangxi University, Nanning 530004, China; 2Advanced Materials Group, Faculty of Engineering, The University of Nottingham, Nottingham NG7 2RD, UK; 3College of Engineering, Mathematics and Physical Sciences, University of Exeter, Exeter EX4 4QF, UK

**Keywords:** water treatment, advanced oxidation, photocatalysis, FeZif, core–shell structure, β-SiC

## Abstract

Herein, we encapsulated modified silicon carbide nanoparticles utilizing a metal–organic backbone. E-SiC-FeZnZIF composites were successfully prepared via Fe doping. The catalysis activity of this bifunctional composite material was evaluated by the degradation of tetracycline (THC) and carbamazepine (CBZ) and the reduction of carbon dioxide (CO_2_). Nano SiC has received widespread attention in advanced oxidation applications, especially in the catalytic activation of peroxymonosulfate (PMS). However, the inferior activity of SiC has severely restricted its practical use. In this study of dual functional composite materials, nano SiC was firstly etched under aqueous alkali. Then, zeolite imidazolate frame-8 (ZIF-8) was used for immobilization. The filling of the etched nano SiC with FeZnZiF was confirmed by SEM, XRD, FTIR, BET, and XPS analyses. In addition, E-SiC-FeZnZIF exhibited excellent catalytic activation of peroxymonosulfate (PMS) to oxidize water pollutants, which can degrade tetracycline hydrochloride (THC), achieving a removal rate of 72% within 60 min. Moreover, E-SiC-FeZnZIF exhibited a relatively high CO_2_ reduction rate with H_2_O. The yields of CO and CH_4_ were 0.085 and 0.509 μmol g^−1^, respectively, after 2 h, which are higher than that of 50 nm of commercial SiC (CO: 0.084 μmol g^−1^; CH_4_: 0.209 μmol g^−1^). This work provides a relatively convenient synthesis path for constructing metal skeleton composites for advanced oxidation and photocatalytic applications. This will have practical significance in protecting water bodies and reducing CO_2_, which are vital not only for maintaining the natural ecological balance and negative feedback regulation, but also for creating a new application carrier based on nano silicon carbide.

## 1. Introduction

While human development has brought about radical changes in human life, it has also caused unprecedented damage to the natural world [1,2]. In particular, the pollution of marine waters and the discharge of various heavy metals, organic pollutants, and medical waste water are growing concerns [3,4]. As a result of the large consumption of antibiotics brought about by the postepidemic era, trace antibiotics have been found in the global ecological environment, raising concerns about the survival status of environmental micro-organisms [5]. This has put a severe strain both on the ecosystem and on human beings themselves, affecting reproduction and the sustainability of future generations [6]. Combined with peak CO_2_ emissions and increased global warming [7], energy and environmental issues are becoming increasingly prominent [7,8].

In recent years, third-generation semiconductors have emerged, including those using silicon carbide, and their applications are flourishing in various areas. With the development of 5G communications, the defense industry, new energy vehicles, and new energy photovoltaics, the demand for silicon carbide is growing at an impressive rate. Silicon carbide is a corrosion- and radiation-resistant material that can withstand high pressures and temperatures. Bedoya-Pinto et al. manufactured an impressive robust ferromagnetic ordered graphene/6H-SiC planar system. They successfully achieved a restricted size Berezinskii–Kosterlitz–Touless phase transition [9]. Alekseev et al. obtained nanoparticles with different surface properties, such as hydroxyl groups and C-C bonds on the surface of a silicon carbide matrix, through KOH and HF treatments [10,11]. This is highly promising for applications related to the degradation of pollutants and reducing carbon dioxide. When utilized as a catalyst, silicon carbide is an ideal substrate and interface for photocatalytic reactions [11]. Therefore, it has shown great potential in environmentally friendly sustainable development applications and new technologies related to energy conversion and storage.

The photocatalytic degradation of water pollutants is an environmentally friendly process because it can effectively convert solar energy into other types of chemical energy. In the past few years, photocatalysts based on metal–organic frameworks (MOFs) have come to prominence. As a result of their polyhedral morphology, large specific surface area, excellent chemical stability, and simple preparation process, they have received significant attention from the scientific community [12,13]. A recent scientific study refined the thermal activation process by developing carbon host ZIF-8 to adsorb Co^2+^ ions [14]. Madsen et al. found short-range disturbances in ZIF glass and applied them to enhance the screening ability of MOF membranes [15]. In addition, current studies show that ligand chemistry in ZIFs has no control over short-range disorder, and the findings reveal structure–property relations that could help to design metal–organic framework glasses [16]. As an expandable porous spider web structure, MOFs have been cleverly combined with metals and organic substances in a bottom-up research approach, which can be used to protect enzymes [17]. Recently, MOFs have been successfully developed and applied in preparing highly active electrochemical water decomposition catalysts due to their tunable nanostructures and excellent porosity [18]. MOFs are also used to store alternatives to traditional fossil fuels, such as hydrogen and methane on ships, which is a considerable challenge as a balance between maintaining ultra-high porosity and the weight- and volume-level surface areas must be established [13,19]. Here, Chen et al. reported a simulated synthesis based on trinuclear metal clusters, proposing a mechanism for converting ZIF-8 into a structure with a polycrystalline form [20]. Erdosy et al. prepared stable dispersions in water by adjusting the internal and external structures of MOF nanocrystals [21]. Li et al. explored the structural evolution and thermal activation process using nitrogen-doped carbon catalysts, and designed a high-performance single Ni site catalyst [22]. As a result, these aqueous fluids can store highly concentrated gases such as oxygen (O^2−^) and carbon dioxide (CO_2_) [23].

Herein, we encapsulated modified silicon carbide nanoparticles by employing a metal–organic backbone. The E-SiC-FeZnZiF composites with a core–shell structure were successfully prepared. The resulting materials exhibit excellent catalytic activation capabilities, which can efficiently separate photogenerated charge carriers, rapidly reduce CO_2_ under light, and catalyze the degradation of potassium persulfate oxidants. The degradation performances of tetracycline hydrochloride (THC) and carbamazepine (CBZ) were explored under various catalytic conditions (catalyst and PMS dosage, THC concentration, and inorganic anion).

## 2. Experimental

### 2.1. Chemicals and Materials

In total, 50 nm β silicon carbide (99.9%), sodium hydroxide (chemical form flake (96%)), ethanol (99% ASC reagent grade), Fe(NO_3_)_3_·9H_2_O (99.9%), 2-methylimidazole (99%), methanol (99.5%), tetracycline hydrochloride (C_22_H_25_ClN_2_O_8_ (USP)), and other conventional reagents, sodium bicarbonate (NaHCO_3_ > 99.5%), sodium hydroxide (NaOH (96%)), sodium thiosulfate (Na_2_S_2_O_3_·5H_2_O (99%)), tetracycline hydrochloride (USP), carbamazepine (99%), and potassium monopersulfate (K_5_H_3_S_4_O_18_ (42~46%)) were purchased from Shanghai McLean Biochemical Technology Co., Ltd. (Shanghai, China).

### 2.2. Preparation of Etched Nano Silicon Carbide

Ten grams of commercially available 50 nm β SiC powder and twenty grams of sodium hydroxide were dissolved in 150 mL DI water. The above prepared solution was heated in water at 90 °C for 2 h. At the same time, the auxiliary magnet was stirred at 400 rpm. The etched nano SiC was collected by centrifugation at 2000 rpm for 10 min. The obtained nano products were washed many times with DI water and ethanol. Lastly, all of the etched nano SiC was dried at 50 °C for 6 h. The principle of the reaction based on the liquid phase can be depicted as in Equation (1).
O_2_ + SiC + 2NaOH + H_2_O→Na_2_SiO_3_ + CO_2_
(1)

### 2.3. Preparation of ZnZiF

First, 3.0 mmol of Zn(NO_3_)_2_·6H_2_O was dissolved in 90 mL of methanol in a beaker [24]. In another beaker, 48 mmol of 2-methylimidazole (2-MI) slowly and uniformly was dissolved in 90 mL of methanol. Then, the solution was slowly dripped from the first beaker into the second beaker. The resulting solution was magnetically stirred at 500 rpm for 15 min and allowed to stand for 24 h. White flocculent gel analogues could be observed at the bottom of the beaker, which were then washed with methanol by centrifugation. The obtained white crystals were dried at 60 °C.

### 2.4. Preparation of E-SiC-ZnFeZiF

Firstly, Fe (NO_3_)_3_ · 9H_2_O was dispersed in 90 mL of methanol, and the solution was continuously stirred with a magnet for 15 min. At the same time, 0.2 g of etched nano SiC was added to the solution. In another beaker, 3.0 mmol of Zn (NO_3_) _2_·6H_2_O and 4 g of 2-MeIm were added to the solution and stirred for 2 h. Furthermore, this was then dissolved in 90 mL of methanol. Thereafter, the contents of the first beaker were poured into the second beaker. The resulting solution was magnetically stirred at 500 rpm for 15 min and allowed to stand for 24 h under natural gravity. E-SiC-ZnFeZiF crystals were obtained, which were then washed with methanol by centrifugation. The obtained black crystals were dried at 60 °C.

### 2.5. Instrumentations

We used a Rigaku D/MAX 2500 V type X-ray diffractometer produced by the Japanese Science Corporation (Tokyo, Japan). The radiation source was Cu target K α X-ray, and the test conditions for all samples were 40 kV and 100 mA. The scanning range was 5° to 80°, and the scanning speed was 5°/min. The FSEM equipment used was a Sigma 300 field emission scanning electron microscope produced by Zeiss AG in Oberkochen, Germany. The prepared catalyst was observed using SE2 and InLens lenses, with a test voltage range of 8 kV. An X-ray photoelectron spectrometer (ESCALAB 250Xi) from the Thermo Fisher Scientific Company in Waltham, MA, USA. the United States was used to characterize the sample’s elemental composition, content, and chemical status. A PLS-SXE 300D (power 300 W, 15 A) xenon lamp light source made by Beijing Porphyry Technology Co. (Beijing, China). was utilized. Advanced oxidation of TC was conducted in a 150 mL glass reactor. In total, 5 mg of commercial SiC and 5 mg of commercial E-SiC-FeZnZiF were dispersed in 100 mL of 10 mg/L of TC solution with ultrasonic treatment for 10 min. The suspension was irradiated under continuous mild stirring using a xenon lamp (Perfectlight PLS-SXE300) with a full spectrum (320–780 nm) at an intensity of 200 mW/cm^2^. In total, 5 mL of the suspension was extracted every 10 min, which was passed through centrifugation at 8000 rpm to reduce the interference of the catalyst. According to the absorbance, the degradation efficiency of THC could be calculated. The catalytic properties of the E-SiC-FeZnZiF were estimated compared with 50 nm of C-SiC according to tetracycline hydrochloride and CBZ degradation. The multi-angle particle size and highly sensitive zeta potential analyzer as manufactured by Brookhaven (model number NanoBrook Omni, manufactured in New York, NY, USA). The fully automatic specific surface area analyzer was manufactured by McMurray-Tick Instruments Ltd. in New York, NY, USA (model TriStar II 3020). The total organic carbon analyzer (model TOC-L CPH) was manufactured by Shimadzu (Kyoto City, Japan).

### 2.6. Organic Pollutant Degradation Measurement

The experiment to assess the catalytic activation degradation of water pollutants was conducted in a beaker at room temperature. The white THC powder particles were ultrasonically dissolved in DI water and stirred for 6 h to prepare a THC solution with a concentration of 20 mg/L and an initial pH of 6.5 (in the contaminant solution, we adjusted the reaction system to neutral by adding dilute hydrochloric acid or sodium hydroxide solution dropwise). A certain amount of catalyst was added to the THC solution and stirred for 5 min to completely disperse the catalyst. Subsequently, a predetermined dose of PMS was added. The reaction solution (1 mL) was sucked out and periodically injected into the Na_2_S_2_O_3_ solution (50 μL) to remove free radicals and prevent further reactions. In total, 5 mL of the suspension was extracted every 10 min, which passed through centrifugation at 8000 rpm to reduce the interference of the catalyst. The overall catalytic degradation efficiency was C_t_/C_0_ (C_t_: concentration of THC at a certain moment of reaction; C_0_: concentration at the beginning of the THC reaction), and the concentration of THC was measured at 350 nm using a UV spectrophotometer.

### 2.7. Photocatalytic Reduction of CO_2_

In total, 10 mg of matrix composites was mixed with 20 mL (0.1 M/L) of sodium bicarbonate solution and put into a 50 mL photocatalytic cell. The resultant solution underwent ultrasonic treatment for 15 min to obtain a colloidal slurry with uniformly dispersed particles. While stirring the ultrasonic sample at a speed of 200 rpm, carbon dioxide gas was introduced. This took 20 min. After the gas had been passed through, the inlet and outlet of the gas were blocked to form a closed space in the tank. Finally, a 300 W, 15 A xenon lamp, with set parameters, irradiated the suspension in a closed glass chamber. At the same time, the illuminated solution in the tank was filtered and collected. The gaseous products extracted from the gas outlet were assessed using a gas chromatograph (Techcomp GC 7900).

## 3. Results and Discussion

### 3.1. Characterization of E-SiC-FeZnZiF

Figure 1a,b show scanning electron microscope images of commercial silicon carbide. The average size of the spherical particles was 50–100 nm. Figure 1c,d show the morphology changes of silicon carbide after etching. They demonstrate that, with the significant decrease in particle size after alkali etching, the nano silicon carbide had a larger specific surface area. Figure 1e,f indicate that the pure ZnZIF bulk particles consist of tiny particles, i.e., below 100 nm. Figure 1g,h indicate that ZnFeZIF forms large crystals with smooth surfaces.

Figure 2a,b show scanning electron microscope images of SiCZnZIF. The average size of the sheets was approximately 1 μm. The SiC in the middle of the interlayer can be clearly observed. According to the EDS diagram of Zn, N, C, and Si, the elements were uniformly distributed, which indicates the successful doping of Zn. The diagrams in Figure 2c,d show that the SiCZnZIF material shrank from a sheet shape to a spherical shape under a 400 °C argon atmosphere. Appendix A shows that the percentage of Zn increased after being burned in the Ar atmosphere. As shown in Figure 2e,f, it was observed that E-SiC was attached to the FeZIF surface. It can be ascertained from the images in Figure 2g,h that SiC-ZnFeZIF bulk crystals were stacked as spherical particles layer by layer. We further studied the E-SiC-ZnFeZIF element composition of the metal matrix composites via EDS element analyses. The results are shown in the color images in Figure 2i. The distribution of N, C, Zn, and Fe in the entire metal matrix composite was uniform, which confirms that E-SiC-Zn-FeZIF successfully doped Zn and Fe. In Appendix A, we measured the particle sizes of C-SiC and E-SiC-ZnFeZIF by zeta potential (pH = 7). The particle sizes of C-SiC, E-SiC, E-SiC-FeZIF, and E-SiC-ZnFeZIF were 6178, 2457, 287, and 322 nm (the average of the three measurements was taken), respectively. In the methanol-as-solvent system, the combined electron micrographs showed that C-SiC and E-SiC agglomerates were evident. In contrast, E-SiC-FeZIF and E-SiC-ZnFeZIF were consistent with the particle sizes observed by SEM. E-SiC had the lowest zeta potential, indicating a greater tendency for agglomeration. E-SiC-ZnFeZIF, on the other hand, had the highest zeta potential, indicating that it was stably dispersed in the methanol system.

Figure 3a shows the XRD patterns of C-SiC and E-SiC. The three sharp peaks are located at 35.73, 60.15, and 71.97°. On the basis of the previously studied ICDD data (JCPDS card (no. 29-1131)), this SiC crystal type appears to be 6-H [25]. In addition, the XRD patterns of E-SiC-FeZIF and E-SiC-ZnFeZIF showed three prominent sharp peaks at 35.50, 59.72, and 71.44°. The corresponding characteristic peaks of ZIF-8 (ZnZIF) were clearly observed in the XRD patterns of the ZIF-8 and E-SiC-ZnFeZIF metal matrix composites, indicating that the introduction of E-SiC nanoparticles into ZnFeZIF did not damage its structure and crystallinity [26]. Figure 3b shows a new intense broad band centered at 861 cm^−1^ for E-SiC [27], which correspond to the Si–C fundamental stretching vibration [28]. For E-SiCFeZIF and E-SiCZnFeZIF, there are corresponding SiC characteristic peaks (861 cm^−1^). This peak does not appear for pure FeZIF, which again demonstrates that the composite composed of etched nano SiC and ZnFeZIF does not destroy the structure of ZnZIF. FeZIF, E-SiC-FeZIF, and E-SiC-ZnFeZIF have peaks at 1383 and 1056 cm^−1^, which obviously correspond to the peaks of nitrate ions and are solid evidence for the construction of a metal–organic framework [29]. C-SiC and E-SiC have similar weak Raman patterns in the peaks in Figure 3c at 1342 and 1602 cm^−1^, respectively. There is no obvious Raman peak for FeZIF. However, four significantly enhanced Raman peaks in the two complexes were observed at 796, 1342, 1602, and 2682 cm^−1^ [30]. In the E-SiCFeZIF and E-SiCZnFeZIF composites in particular, the characteristic peaks of silicon carbide, i.e., 1342 and 1602 cm^−1^, were significantly enhanced, demonstrating a strong interaction between E-SiC and ZnFeZIF [31].

### 3.2. Organic Pollutant Degradation Performance

#### 3.2.1. Catalytic Performance in Degradation of Tetracycline Hydrochloride and CBZ

The catalytic performance of E-SiC-ZnFeZIF was evaluated in detail by changing the conditions that affect various parameters during the catalytic degradation of CBZ and THC. As shown in Figure 4a–c, CBZ and THC degradation (3–8%) with the SiC and PMS catalysts was negligible. We supplemented the experiments with degradation of various catalysts by sonication for 3 min, followed by no stirring, in Appendix A. In the degradation experiments without stirring, C-SiC and E-SiC were degraded (including adsorption) by 7% and 9%, respectively, within 60 min. The degradation efficiencies of E-SiC-FeZIF and E-SiC-ZnFeZIF were 38% and 54%. In contrast, degradation efficiencies of 18%, 18%, 50%, and 72% were observed for C-SiC, E-SiC, E-SiC-FeZIF, and E-SiC-ZnFeZIF, respectively, in the degradation experiments with stirring. Significantly lower degradation efficiencies were observed without stirring. This indicates that stirring accelerates the contact between the catalyst surface and the organic contaminants, promoting both free radical and non-free radical pathways. The effect of stirring increases the contact area and interaction between the components in the solution, facilitating the chemical reaction and speeding up the reaction rate. It allows the reactants to mix uniformly and facilitates the reaction and heat dissipation. This promotes the degradation of organic pollutants (THC). In addition, the adsorption ability of E-SiC-ZnFeZIF to CBZ and THC molecules is poor. However, almost 72% of the THC and 18% of the CBZ were degraded after 60 min when the E-SiC-ZnFeZIF (metal matrix composite) was used as a catalyst for a wastewater solution in the presence of PMS. The highest total organic carbon removal by degradation of THC (20 ppm 100 mL) was 65%. With THC concentrations of 10, 20, and 40 ppm, the degradation efficiency was 57%, 48%, and 38%, respectively. E-SiC-ZnFeZIF is composed of SiC and ZnFeZIF, which is essential to study the catalytic activities of these substances. Almost no CBZ and THC were degraded when C-SiC and E-SiC were used as a catalyst in the presence of PMS. By comparing the degradation efficiency of E-SiC and E-SiC-ZnFeZIF, with the analysis of specific surface area in the previous section, it is seen that the composite material has a higher catalytic degradation performance than either material alone, which indicates a clear interaction between the two materials. Therefore, the high catalytic activity is mainly due to the activation of ZnFeZIF. However, there must be relevant synergies between the E-SiC and ZnFeZIF, because E-SiC-ZnFeZIF exhibited a higher degradation activity than pure ZnFeZIF. As E-SiC-ZnFeZIF is composed of Fe^2+^ and Fe^3+^ based on ZnFeZIF, the above substances can react with PMS to generate sulfate and persulfate radicals, as shown below (Equations (2)–(4)).
Fe(III) + HSO_5_^−^ → Fe(II) + SO_5_^•−^ + H^+^
(2)
Fe(II) + HSO_5_^−^ → Fe(III) + SO_4_^•−^ + OH^−^
(3)
2SO_5_^•−^ → 2SO_4_^•−^ + O_2_
(4)

The experimental results show that the E-SiC-ZnFeZIF/PMS integrated system is fast and efficient in the mineralization of THC, confirming that E-SiC-Zn-FeZIF has high catalytic degradation activity. According to the experimental results, most of the tetracycline hydrochloride can be degraded and removed in the E-SiC-ZnFeZIF system. Therefore, we continued to explore the effects of various environmental and internal factors on the degradation of tetracycline hydrochloride. As shown in Figure 4d, when the PMS was increased to 1 mM, the degradation efficiency reached the maximum value (52% is the same as 0.5 mM). When the concentration of PMS in the degradation environment was 2 mM, the degradation efficiency slightly decreased compared to the aforementioned example. A high concentration of PMS increases the yield of reactive species in the system. On the one hand, a large number of reactive species will undergo radical annihilation reactions, reducing the efficiency of THC removal. On the other hand, an increase in the yield of reactive species also increases the chance of reaction with SiCZnFeZIF in the system and reduces the conversion efficiency between Fe^3+^ and Fe^2+^ ions. The activation process is inhibited, thereby indirectly affecting the degradation of the pollutants. When the amount of catalyst (E-SiC-ZnFeZIF) was increased from a lower quantity (from 5 mg/mL to 50 mg/mL), the overall degradation efficiency decreased from 72% to 25% (Figure 4e). This suggests that in a high-concentration catalyst environment, the free radicals used for degradation may undergo self-quenching (Equations (5) and (6)), which becomes the main factor limiting the increase in the reaction rate [32].
SO^•4−^ + SO^•4−^ → S_2_O_2_ ^8−^
(5)
SO_4_^−•^+ OH^•^ → HSO_5_^−^
(6)

As organic wastewater is discharged into the natural ecological environment, over time and under the influence of external factors, the inorganic anions generated in the water body will compete with organic substances, or even react with active substances, resulting in the loss of their activity. This seriously interferes with the catalytic degradation process [33]. However, when phosphate ions and bicarbonate ions exist, as in Figure 4f, where their concentration is 10.0 mmol/L, they significantly reduce the degradation efficiency. For example, when 10.0 mmol/L of phosphate ions or bicarbonate ions is present, the THC degradation efficiency is 17% and 15%, respectively. Through the simultaneous monitoring of the pH change in the solution, it can be seen that the introduction of phosphate ions or bicarbonate ions can increase the pH of the solution, i.e., to 10.1 and 8.5, respectively. Phosphate and bicarbonate ions added to the system make the solution alkaline, weakening the interaction between E-SiC-ZnFeZIF and PMS, thus reducing the production of active species. By the analysis of the BET specific surface area (Figure 4g,h), the adsorption–desorption curves in the nitrogen environment, the adsorption of the composite E-SiC-ZnFeZIF is 47 cm³/g STP (relative pressure (P/Po)), which is much smaller than the 180 cm³/g STP (relative pressure (P/Po)) of E-SiC. The specific surface area of the composite E-SiC-ZnFeZIF is 17.22 m²/g, with a minimum pore volume of 0.079 cm³/g (<1. 663.1 Å diameter at P/Po = 0.988) and 0.075 cm^3^/g for pore widths in the range 60–100 nm, which is smaller than the 0.25 cm^3^/g of E-SiC. The percentage of pore widths in the range 0–60 nm is much smaller than in the region of E-SiC (based on a comparison of the triangular regions), so its adsorption capacity can be ignored. Furthermore, the maximum pore volume and specific surface area of E-SiC has a less than 10% adsorption degradation capacity over 60 min, which proves that the adsorption capacity of the composite is the weakest. The smaller the grain size of the catalyst, the larger the specific surface area and the rougher the surface, both of which are more conducive to the adsorption of organic pollutants. However, if the composite grain size becomes larger, the crystal shape is more complete. Then, the adsorption performance decreases, however the degradation performance increases. This point proves that the catalytic activity is dominated by the complex which formed from the two factors (E-SiC and ZnFeZIF) and the interaction of the composites is more significant. Adsorption performance is a secondary factor for degradation. However, some chloride ion salts that can produce Cl also have a weaker oxidation ability. For example, (E (Cl/Cl^−^) = 2.43 V, E (Cl^2−^/Cl^−^) = 2.13 V) (Equations (7) and (8)) [34]. Therefore, in the water model of the catalytic activation degradation of THC, chloride ions weakened the E-SiC-ZnFeZIF/PMS system to a certain extent [33,35].
SO_4_^−^ + Cl^−^ → SO_4_^2−^ + ∙Cl (7)
Cl^−^ + Cl^−^ → 2∙Cl^2−^
(8)

Previous research reports found that low concentrations of NO_3_^−^ had a significant inhibitory effect on THC degradation. The corresponding research mechanism can be described as the reaction of NO_3_ with SO_4_^•^ and OH^•^, resulting in a low activity of NO_3_^•^ and NO_2_^•^ in water (Equations (9)–(11)) [36]. There is also a mechanism to explain the inhibition of THC degradation by Cl^−^. The Cl^−^ present in water pollutants can combine with free radicals to form chlorine free radicals with a low catalytic activity, such as Cl^•^ and Cl_2_^•−^ (Equations (12) and (13)) [37]. Therefore, before applying the composite material for degradation, the phosphate ion, bicarbonate ion, and chloride ion values in the wastewater should be detected, and the addition of pretreatment measures should be considered. In a single PMS or E-SiCZnFeZIF degradation solution system, the degradation efficiency is only 46.67 and 7.23%, respectively, as shown in Figure 5a. The synergistic effect of PMS and E-SiCZnFeZIF was demonstrated. In the subsequent mechanism analysis, we proposed the relevant reaction process. Then, we tested the stability of the composite. After three cycles, the degradation performance of the composite decreased to 40.59%. Using the dissolution data of the leaching solution for 1 h, it was shown that the stability of ZIF materials in water and the crystal structure were negatively affected, leading to the precipitation of Fe^3+^. According to the integrated wastewater discharge standard (GB8978-1996), the water quality standard for sewage discharged into urban sewers is 10 mg/L. On the basis of the ICP measurements from Figure 5c, the Fe content in the leaching solutions for 0.5 h, 1 h, and 20 h is 2.001, 4.165, and 10.898 mg/L, respectively. These values are lower than the national standard within 1 h and are suitable for use in widely degrading water pollutants. The decrease in degradation properties indicates the instability of this ZIF composite in aqueous solution, combined with leaching experiments. We know that the structure of the metal–organic skeleton is damaged by immersion in aqueous solutions for longer than 20 h. The leaching of Fe^3+^ ions within 1 h is less. Therefore, we should filter the catalyst out of the solution in time. In Figure 5d, we conducted free radical capture experiments with 10 mM of p-benzoquinone, tert-butyraldehyde, and EDTA-2Na. The three types of capture agents have inhibitory effects on the degradation of tetracycline hydrochloride by composite-material-catalyzed activated PMS. O_2_^•−^ and photogenerated holes (h^+^) play a leading role, while SO_4_^•−^ and OH^•^ are both involved in the degradation of tetracycline hydrochloride. SiC and Fe-ZIFs are the main active centers. The doping of Fe elements makes the catalyst’s ability to transfer electrons more stable while accelerating the redox cycle of Fe^3+^ and Fe^2+^, enabling the catalyst to efficiently catalyze the decomposition of PMS to produce active species (SO_4_^•−^, OH^•^).
NO_3_^—^ + SO_4_^•−^ → SO_4_^2−^ + NO_3_
(9)
NO_3_^—^ +OH^•^ → OH^—^ + NO_3_
(10)
NO_3_^—^ + H_2_O + e^−^ → 2OH^—^ + NO_2_
(11)
Cl^—^ + SO_4_^•−^ → SO_4_^2−^ + ∙Cl (12)
Cl^—^ + Cl^•^ → 2Cl^—^(13)

#### 3.2.2. Organic Pollutant Degradation Mechanism

According to the summary and the description above, the mechanism and acquisition method for the degradation of THC by SO_4_^−^ are shown in Equations (1)–(3). Then, SO_4_^−^ becomes OH^−^ under a certain action (Equation (14)) [38].
SO_4_^2−^ + H_2_O → SO_4_^2−^ + OH^•^ + H^+^
(14)

On the basis of the previous analysis and discussion, this study proposes a corresponding E-SiC-ZnFeZIF activation mechanism, as shown in Figure 5. First, HSO_5_^−^ ions present on the surface of E-SiC-ZnFeZIF can be coupled to Fe (II) species to produce SO_4_ free radicals (Equations (2) and (3)). SO4^−^ free radicals are oxidized to OH^•^ by water molecules. Moreover, the electrons transferred from HSO_5_^−^ to Fe (III) ensure the production of SO_5_ through Equations (3) and (4). In addition, SO_4_^−^ can also be produced by subsequent SO_5_^−^ conversion (Equation (6)). Therefore, the charge balance of the entire system is maintained. The free radicals generated in the cycle can mineralize and degrade THC into small molecules, and even inorganic substances.

In order to compare the performance of different composites for the photodegradation of pollutants in detail, the advantages and disadvantages of E-SiCZnFeZIF, along with silicate composites and silica-based composites, are listed in Appendix A. This shows that the E-SiCZnFeZIF-catalyzed activation of potassium persulfate for pollutant degradation has a broad range of applications, without the need for light and with a low catalyst dosage. In Appendix A, we compare the performance of composite catalytically activated PMS systems in Fenton-like systems for the degradation of pollutants. It was found that relatively good degradation can be achieved with minor SiC input. The modified silicon carbide and ZnFeZIF have strong interactions, which can be inferred by comparing different catalyst degradation experiments, free radical trapping experiments, specific surface area analysis, and zeta potential analysis. The composites have the smallest specific surface area and the smallest pore volume. However, their catalytic degradation performance and photocatalytic reduction performance are the best. We propose a possible mechanism: the interaction of the silicon carbide and ZIF materials enhances the cycling of Fe^2+^ and Fe^3+^, enhancing the generation of free radicals (Figure 6). In contrast, silicon carbide nanoparticles, as an inert semiconductor material, are ideal as an additive phase and carrier. In combination with ZnFeZIF, the zeta potential values become larger and more stable in systems where methanol is used as a solvent compared to C-SiC. It can therefore be applied in aqueous environments with a high proportion of organic systems for the degradation of pollutants.

### 3.3. CO_2_ Reduction Performance

As demonstrated in Figure 7a,b, this experiment was carried out using the liquid-phase method, and the gas in the volumetric flask was collected for testing. In 0.5 h, the carbon dioxide reduction capacity of the E-SiC was stronger than that of the original sample. The yields of CH_4_ and CO were 1.69 and 1.3 times as much as before. Therefore, in this section, we discuss the selectivity of the reduction products. The selectivity of the etched silicon carbide for methane increased by 6% (to 76.0%) compared to C-SiC. Moreover, after 1 h, the selectivity to methane decreased to 67.3%, and the CH_4_ and CO yields of the E-SiC were 0.249 and 0.121 μ mmolg^−1^ h^−1^, respectively. However, after 1 h, the sample yields of CH_4_ and CO after etching were higher than that of the original sample. Figure 7c,d show that the CH_4_ and CO production of FeZIF doped with E-SiC was significantly higher than that of C-SiC within half an hour. After E-SiC was doped with FeZIF and FeZnZIF, the selectivity of the composite sample to CO improved. This shows that the sample had a higher selectivity for CO after being combined with E-SiC. Overall, the addition of Zn reduced the production of CH_4_ and CO over half an hour, as shown in Figure 7d.

When tested in the dark, no products were detected, which indicates that the CO_2_ reduction reaction was photoexcited (Appendix A). As Figure 8a,b show, after 1 h of xenon lamp irradiation, the CO and CH_4_ yields obtained for E-SiC-ZnFeZIF were 0.085 and 0.509 μmolg^−1^, respectively, which were higher than the production of the original C-SiC (CO: 0.084 μmolg^−1^, CH_4_: 0.209 μmolg^−1^). Interestingly, as Figure 8c,d show, the E-SiC-FeZIF (CO: 0.043 μmol g^−1^, CH_4_: 0.294 μmol g^−1^) also exhibited higher CO and CH_4_ yields than C-SiC under 1 h of xenon lamp irradiation, which indicates that Fe-doped SiC composites play a key role in promoting CO_2_ reduction. The highest CO_2_ reduction rate was obtained in the presence of E-SiC-ZnFeZIF, and the corresponding yields of CO and CH_4_ were 0.085μ mol g^−1^ and 0.509 μ mol g^−1^, respectively, for 1 h of irradiation. The CH_4_ yield was 2.43 times that of the original C-SiC. Compared with C-SiC, the selectivity of etched silicon carbide for CH_4_ increased by 15% (to 85.7%). The yield of carbon monoxide was 24.3%. As shown in Appendix A, the total yield of the composite is more than twice as high as that of C-SiC. The conversion to methane increased from 71.4% to 85.7%. The conversion to CO increased from 28.6% to 14.3%. This again demonstrates the strong synergy between SiC and ZnFeZIF. E-SiC-ZnFeZIF produces more reactive intermediates in the photocatalytic reduction of CO_2_ (such as monodentate, bidentate, and acetate follow-on reactive intermediates) than the single C-SiC. Finally, in Appendix A, we list the performance of silicon carbide and its composites for the photocatalytic reduction of CO_2_ as reported in other research literature. It shows a twofold improvement in the CH_4_ performance of E-SiCZnFeZIF compared to 50 nm of C-SiC. Appendix A shows the leaching concentration of Fe^3+^ ions in the carbon dioxide reduction experiment. The leaching concentrations were 3.022, 7.053, and 13.578 ppm at 0.5, 1, and 20 h, respectively. The increase in the leaching concentration of metal ions demonstrates the destruction of the metal–organic skeleton’s structure under the xenon lamp irradiation. Therefore, we should further reduce the leaching of Fe^3+^ ions by stabilizing the structure of the composite. The leaching of Fe^3+^ ions within 1 h is less. This may be able to be reduced by filtering the catalyst out of the solution after 1 h.

### 3.4. Optical Property and Energy Band Structures

The XPS survey spectrum of Si2p and the C1s XPS exemplary spectrum of C-SiC and E-SiC-ZnFeZIF are shown in Figure 9a–h. From the whole XPS spectrum (Figure 9a–c), it is found that the main elements are C, Si, and O. The peak positions of Si2p and C1s are 101.23 and 284.85 ev, respectively [39]. Figure 9d–f show the XPS survey spectrum of E-SiC-ZnFeZIF and the XPS fine spectra of Si2p, C1s, Zn2p, and Fe2p. The peak positions of Si2p and C1s are 100.70 and 284.20 ev, respectively [40]. In addition, two new, relatively weak peaks appear in C1s, with binding energies of 288.35 and 282.35 ev [41]. In addition, BE decreases for these peaks by 0.53 and 0.65 ev, respectively. This blue shift shows that the relative interaction between Fe, Zn, and S and C was enhanced. The doping of Fe was successfully verified. In addition, two relatively new weak peaks appear in C1s, with binding energies of 288.35 and 282.35 ev. They represent a C=O bond and Si=C bond. In the Zn 2p core-level spectrum of E-SiC-ZnFeZIF (Figure 9g), two significant peaks at 1044.5 and 1021.9 eV are observed, corresponding to the binding energies of Zn 2p1/2 and Zn 2p3/2 derived from ZIF-8, respectively. The spectrum of Fe2p is broken down into three distinct peaks. Three major peaks are detected at 711.6 (Fe^2+^), 718.2 (Fe^3+^), and 723.3 eV (Fe^3+^), in agreement with previous reports [42]. This also indicates the successful preparation of the E-SiC-ZnFeZIF composite. The binding energy of the C1s of E-SiC-ZnFeZIF was transferred to a lower binding energy, which shows that the E-SiC was closely decorated on the surface of ZnFeZIF via a strong interaction [31,43]. On the basis of the above analysis, we can conclude that the E-SiC tightly covers the surface of FeZnZIF without changing the phase structure, which verifies the successful manufacturing of E-SiC-ZnFeZIF.

In PL testing, the excitation luminescence is 250 nm. There are two obvious emission peak positions at 350 and 395 nm in Figure 10a. The order of luminescence intensity of the samples is E-SiC-ZnFeZIF > FeZIF > E-SiC. The combination of photogenerated electrons and holes in E-SiC-ZnFeZIF generates the most photons, resulting in the strongest intensity of photoluminescence. As displayed in Figure 10b, E-SiC-ZnFeZIF has a higher charge transfer resistance at the interface with the electrolyte. Figure 10c shows that the relative photocurrent density of the SiC sample compounded with Zn and Fe is lower than the original sample. This may be because the outer layer of silicon carbide is wrapped with a thicker ZnFeZIF material [44]. The thicker organic skeleton shell will prevent the transfer of photogenerated electrons, thereby reducing its overall photocurrent response value [45]. Figure 10d shows the cycling yield for CH_4_ and CO production of E-SiC-ZnFeZIF. The CO_2_ reduction tests were repeated five times. After the second cycle, subsequent cycles have increasingly lower yields. In conjunction with the previous leaching experiments, we can infer that the cyclic stability of the composite is less. This is attributed to the destruction of the ZIF material by the aqueous system. Based on the reproducible conditions of the two experiments, previous literature research, and related leaching experiments, we know that ZIF materials are unstable in aqueous environments [46]. Some literature studies indicate that the stability of ZIF materials in water can be improved by compounding with Zn elements. Therefore, we have made E-SiC-ZnFeZIF in our experiments, and we hypothesize that by changing the ZnFe ratio, we can enhance the stability of the composite and reduce the leaching of Fe^3+^ ions [47]. The results show that ZIF-8 crystallites are not stable in water under ambient conditions [48]. The amount of ZIF-8 crystallites that are dissolved depends on the ZIF-8/water ratio and contact time. Solid samples collected by filtration may contain residual ZIF-8, which is responsible for the confusion in the literature about the water stability of ZIF-8 [49].

The light trapping ability of each material was investigated by UV–vis–NIR diffuse reflectance spectroscopy (Figure 11a). E-SiC nanoparticles and E-SiC-ZnFeZIF have similar absorption properties. The latter has a higher absorption peak at 400–600 nm, which further confirms the existence of FeZIF in the composite material. The main absorption edges of both materials are at 400 nm and the absorption region is 400–1100 nm. In particular, the absorption is strong in the near infrared region. This is due to the presence of abundant oxygen vacancies on the surface of both materials, which produce a metal-like LSPR effect [50]. FeZIF, on the other hand, absorbs more strongly in the 400–700 nm region. The absorption of FeZIF is weaker in the near infrared region. Figure 11b shows the band gaps for SiC, FeZIF, and E-SiC-ZnFeZIF at 1.92, 1.50, and 1.15 eV, respectively.

In situ FTIR was performed to study the groups formed on the surface of the E-SiC-ZnFeZIF. This was used to determine the intermediates formed in the CO_2_ reduction. As shown in Figure 12a,b, in the presence of CO_2_ and water there are no obvious peaks at 10 min with the light off and 50 min with the light on. This indicates that no obvious intermediates were formed. In the tests of E-SiC-ZnFeZIF, bicarbonate (HCO_3_*, 1490 cm^−1^) [51], a CO* active substance (2076 cm^−1^) [52], carboxylate (COOH*, 1243 and 1533 cm^−1^) [53], monodentate (m-CO_3_, 1280 cm^−1^ and 1430 cm^−1^) [54], and bidentate carbonate (b-CO_3_ 1630 cm^−1^) [55] were identified. The peaks of the corresponding intermediates also appear without the xenon lamp on, indicating that E-SiC-ZnFeZIF is sensitive to infrared light sources and can absorb energy from near-infrared light sources, leading to the onset of the CO_2_ reduction process and the production of the corresponding intermediates.

In CO_2_ reduction, there are three main types of free radicals that play a role: the hydroxyl radical (•OH), superoxide radical (O_2_^•−^), and electrons (e^−^). •OH is one of the strongest oxidants and can be generated by the photolysis of water [56]. In the photocatalytic CO_2_ reduction process, •OH can react with CO_2_ to produce carbonyl compounds, formic acid, methanol, and other reduced products. O^2−•^ usually works in coordination with •OH to exert synergistic oxidation effects. O^2−•^ can be generated by the photolysis of oxygen and can cooperate with •OH to produce oxidized products such as hydrogen peroxide (H_2_O_2_) [57]. The electron (e^−^) is an intermediate in the reduction reaction. Electrons can be directly transferred to CO_2_ to produce reduced products such as formaldehyde and alcohols. In addition, electrons can also react with O^2−•^ and •OH to eliminate these free radicals, preventing unnecessary oxidation reactions caused by them. This study provides a comprehensive understanding of the mechanisms involved in the photocatalytic CO_2_ reduction process and highlights the importance of free radicals in this reaction. The nano-polyaniline electrode in this solution shows low resistance and low charge transfer impedance (R-ct), which leads to a rather high reduction current density (−16.8 mA cm^−2^) and low overpotential (−0.44 V vs. Ag/AgCl) [58]. This solution is environment-friendly and mimics that in photosynthesis. 

In the literature, the majority of related studies calculate the process of carbon dioxide intermediates in detail through a DFT simulation and in situ infrared spectroscopy. We based our approach on the research of Li et al. and Ramis et al. The reaction intermediates in the CO_2_ reduction process were detected by Fourier transform infrared spectroscopy (FTIR). Peaks at 1619 and 1620 cm^−1^ were detected, and these peak intensities gradually increased with the prolongation of the irradiation time. Both peaks belong to the COOH * group, which is usually considered a key intermediate in the process of CO_2_ reduction to CO or CH_4_. The proposed mechanism is based on the free radical capture experiments, and possible reaction paths are depicted in Figure 13a,b. This test was carried out with the Bruker A300 to capture the signal of hydroxyl radicals. There was no signal at 0 min and a strong hydroxyl peak signal appeared at 10 min (Figure 13a). This indicates that the radiation generated by an electron leaps to the surface under light, which leads to the production of hydroxyl radicals. Thus, it is involved in the process of carbon dioxide reduction. Detailed experimental parameters are given in Appendix A. The second mechanism denotes reducing the carbonate in further continuous steps to generate formaldehyde and possibly methanol. It has been proved that hydroxyl radicals, photogenerated holes, and superoxide radicals are the main reactive active substances. In Figure 10c, the first reaction is the photoreduction of molecular carbon dioxide to formic acid, which can be further reduced to formaldehyde and methanol in a strictly continuous path, or can be evolved into gaseous products through photoreforming, i.e., the accumulated organic molecules (HCOOH, HCHO, and CH_3_OH) are oxidized through photogenerated pores to generate H_2_ + CO/CO_2_. In Figure 14, we propose the mechanism of photocatalytic reduction of CO_2_ into CO and CH_4_ by the E-SiC-ZnFeZIF.
∗ + CO_2_ + e− + H+ → COOH∗ (15)
COOH∗ + e− + H+ → CO∗ + H_2_O (16)
CO∗ + e− + H+ → CHO∗ or CO∗ → CO↑ + ∗ (17)
CHO∗ + e− + H+ → CH_2_O∗ (18)
CH_2_O∗ + e− + H+ → CH_3_O∗ (19)
CH_3_O∗ + e− + H+ → CH_4_↑ + O∗ (20)
O∗ + e− + H+ → OH∗ (21)
OH∗ + e− + H+ → H_2_O + ∗ (22)

## 4. Conclusions

In summary, the E-SiC-ZnFe material supported by ZIF was successfully prepared by doping with Fe, which was first well characterized. Paralleled with C-SiC and ZnZIF, the E-SiC-ZnFeZIF metal matrix composites showed excellent catalytic activity when activating PMS to degrade THC. The catalytic degradation capability of THC is obviously related to the dose of different species (E-SiC-ZnFeZIF, PMS, phosphate ion compound, bicarbonate ion compound, and chloride). SO_4_^−^ free radicals obviously play a major role in attacking THC. In summary, this study provides a silicon carbide-doped metal matrix composite catalyst for degrading organic compounds in wastewater. In addition, under irradiation from a xenon lamp, the semiconductor photocatalysis material has the ability to reduce carbon dioxide and has a higher yield of methane and carbon monoxide. This material has great potential in addressing energy shortage problems and environmental pollution. Moreover, it sets a precedent for applying third-generation semiconductors (SiC) in advanced oxidation and carbon dioxide reduction.

## Figures and Tables

**Figure 1 nanomaterials-13-01664-f001:**
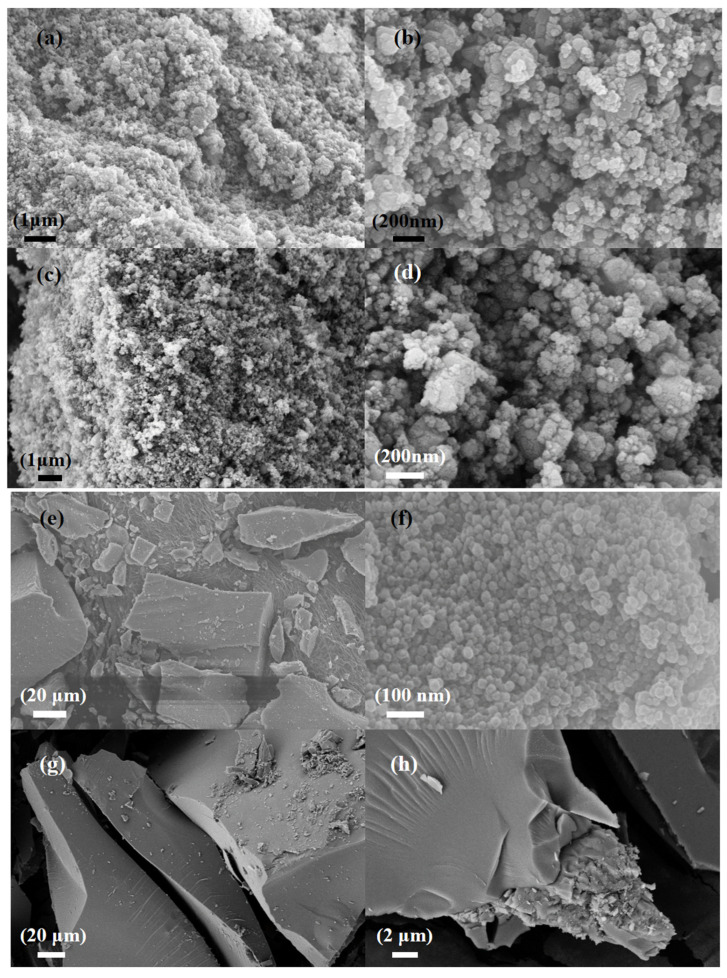
SEM images of (**a**,**b**) the commercial SiC (C-SiC), (**c**,**d**) the etched SiC (E-SiC), (**e**,**f**) the ZnZIF, and (**g**,**h**) the ZnFeZIF.

**Figure 2 nanomaterials-13-01664-f002:**
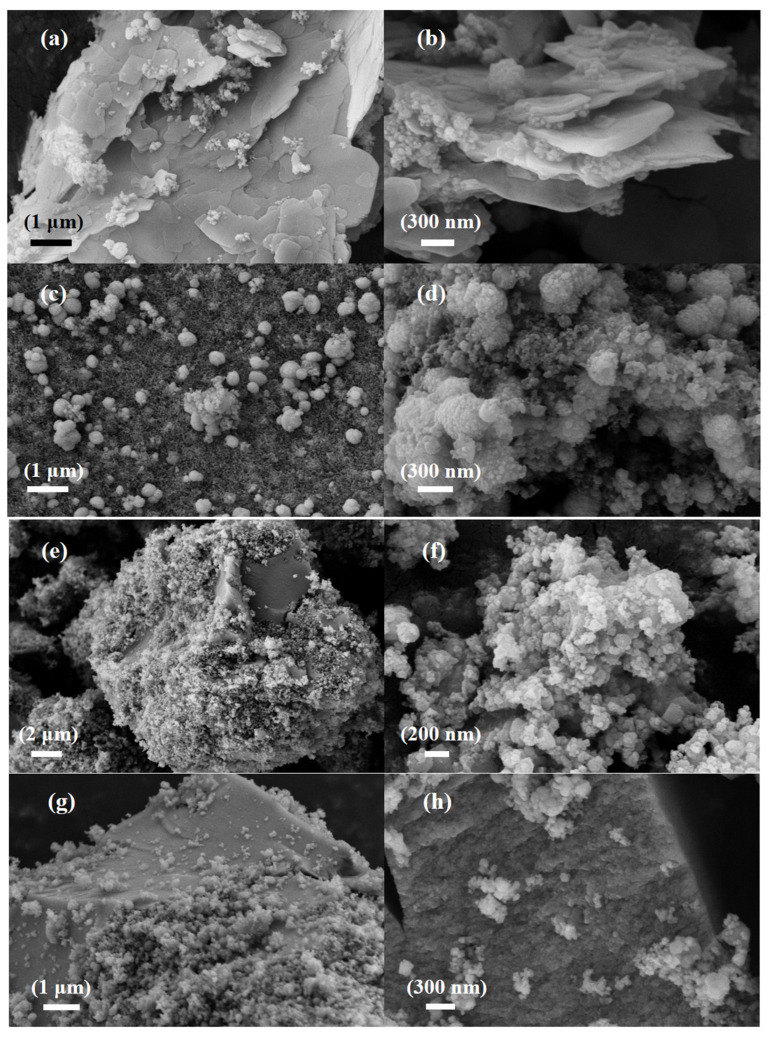
SEM images of (**a**,**b**) the E-SiC-ZnZIF, (**c**,**d**) the SiC-ZnZIF400Ar, (**e**,**f**) the E-SiC-FeZIF, and (**g**,**h**) the E-SiC-ZnFeZIF. (**i**) EDS image and element mapping of E-SiC-ZnFeZIF.

**Figure 3 nanomaterials-13-01664-f003:**
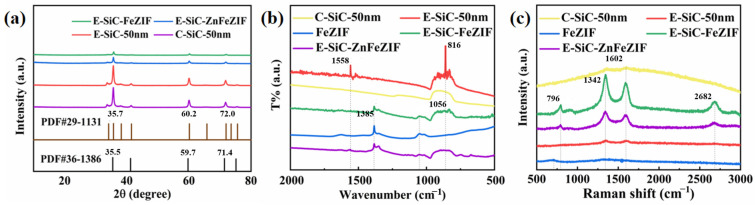
(**a**) XRD, (**b**) FTIR, and (**c**) Raman patterns of the obtained samples, including C-SiC, E-SiC, E-SiC-FeZIF, and E-SiC-ZnFeZIF.

**Figure 4 nanomaterials-13-01664-f004:**
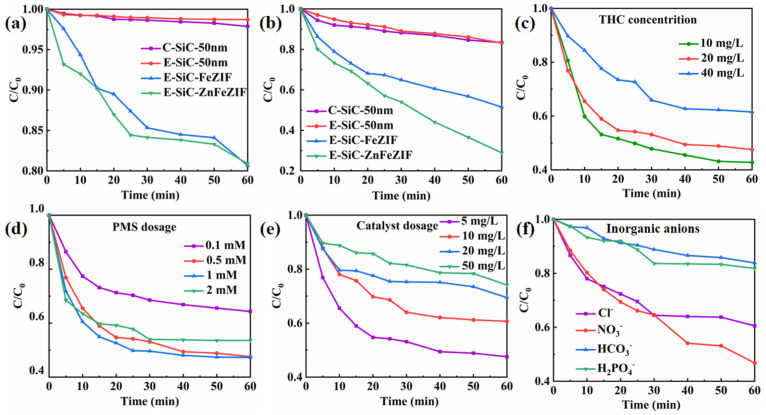
(**a**,**b**) Show the degradation performance of C-SiC, E-SiC, E-SiC-FeZIF, and E-SiC-ZnFeZIF for CBZ and THC in the PMS system. The effect of various conditions on the THC removal rate: (**c**) initial concentration of THC; (**d**) PMS concentration; (**e**) catalyst dosage; (**f**) effect of inorganic anions (experimental conditions: [CBZ] = 20 mg/L, [PMS] = 0.5 mM, [E-SiC-ZnFeZIF] = 50 mg/L, pH = 6.5, T = 25 °C, when an experimental environment parameter was changed, the remaining influencing parameters remained unchanged, ensuring a single experimental variable). (**g**) Plot of the adsorption–desorption curve of the photocatalyst. (**h**) Porosity distribution by NLDFT.

**Figure 5 nanomaterials-13-01664-f005:**
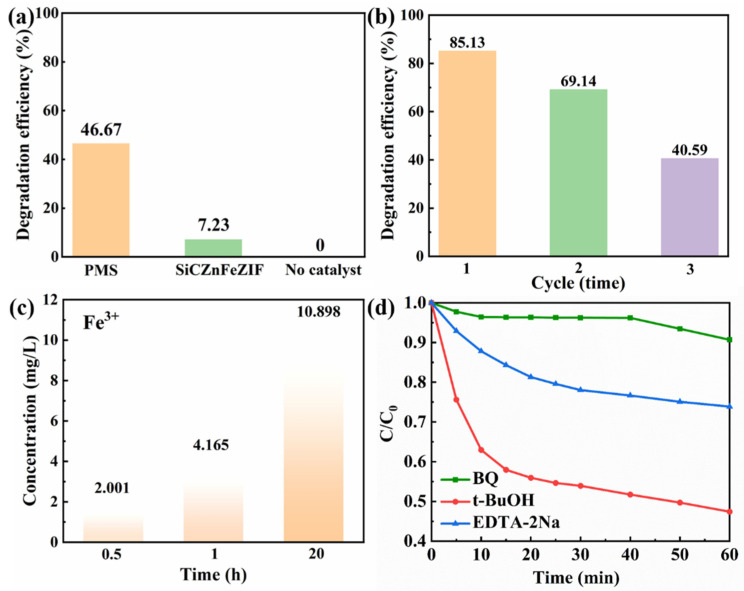
(**a**) Pharmaceutical removal in the absence of a catalyst and in the absence of PMS or E-SiCZnFeZIF; (**b**) degradation cycle experiment; (**c**) metal (Fe^3+^) leaching in the liquid phase after degradation; (**d**) free radical trapping experiments conducted with 10 mM of p-benzoquinone, tert-butyraldehyde, and EDTA-2Na (experimental conditions: [THC] = 10 mg/L, [PMS] = 0.5 mM, [E-SiC-ZnFeZIF] = 50 mg/L, pH = 6.5, T = 25 °C, when an experimental environment parameter was changed, the remaining influencing parameters remained unchanged, ensuring a single experimental variable).

**Figure 6 nanomaterials-13-01664-f006:**
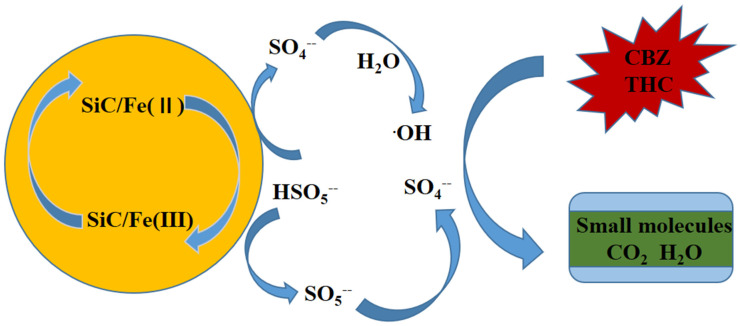
The reaction mechanism of the catalytic activation of the ESiC-ZnFeZIF/PMS system.

**Figure 7 nanomaterials-13-01664-f007:**
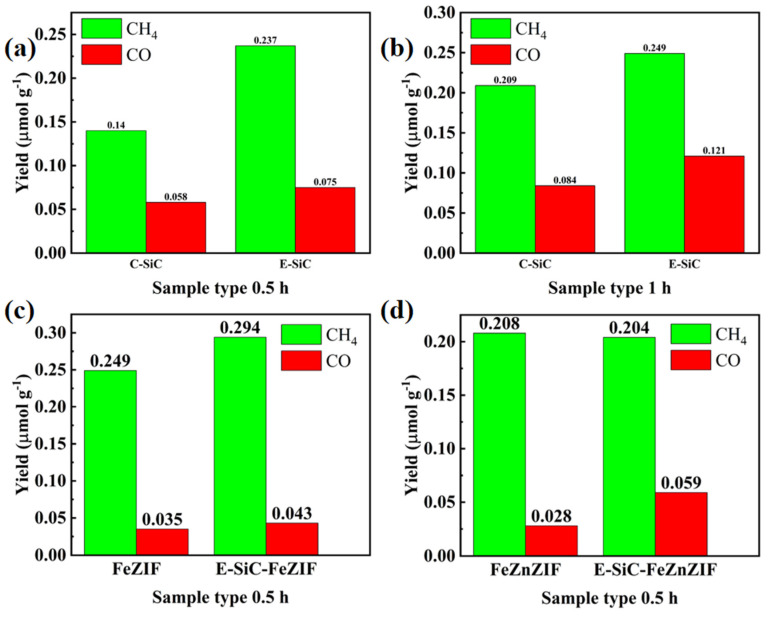
Photocatalytic CO_2_ reduction performances. (**a**–**d**) Show the CO and CH_4_ production rates of the obtained samples under the xenon light for 0.5 and 1 h.

**Figure 8 nanomaterials-13-01664-f008:**
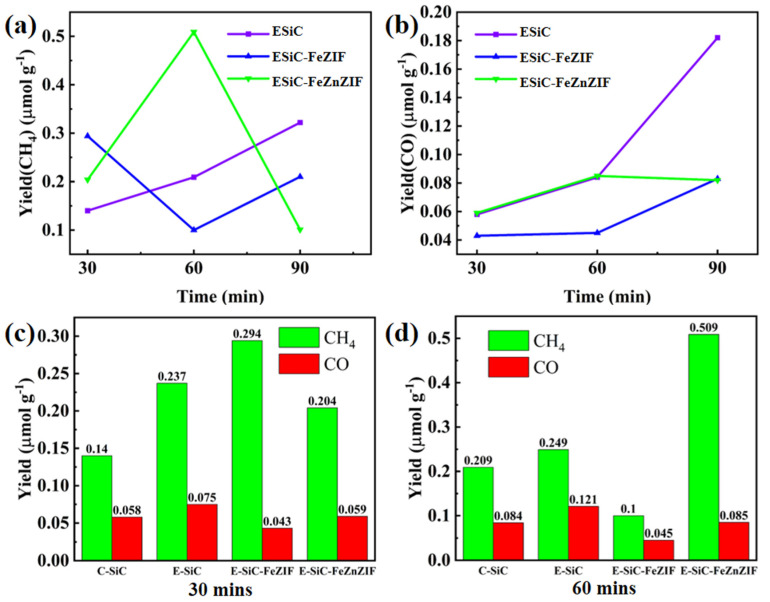
The photocatalytic CO_2_ reduction production. (**a**,**b**) Show the CO and CH_4_ production rates of E-SiC, E-SiC-FeZIF, and E-SiC-FeZnZIF under the xenon light for 30 min to 90 min. (**c**,**d**) Show the CO and CH_4_ production rates of all samples under the xenon light for 0.5 and 1 h.

**Figure 9 nanomaterials-13-01664-f009:**
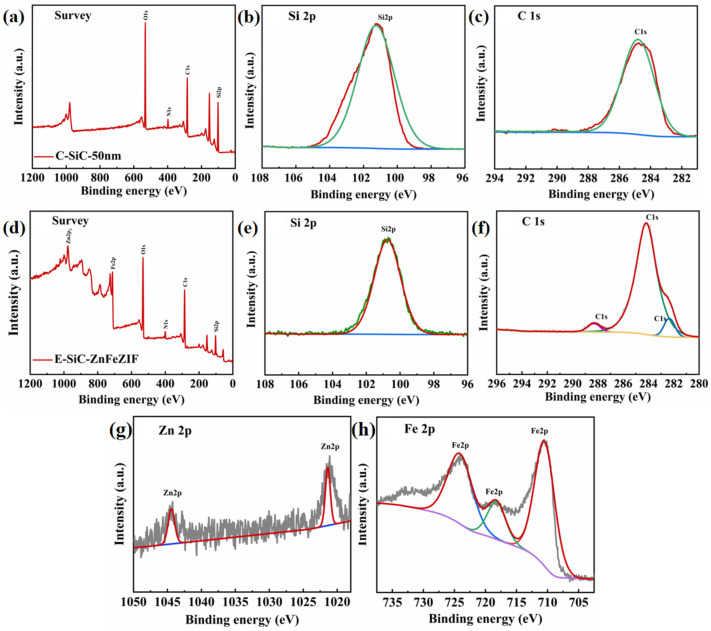
(**a**) XPS spectra of C-SiC’s survey, (**b**) Si 2p XPS spectra, (**c**) C 1s XPS spectra, (**d**) XPS spectra of E-SiC-ZnFeZIF’s survey, (**e**) Si 2p signals of E-SiC-ZnFeZIF, (**f**) C 1s XPS spectra of E-SiC-ZnFeZIF, (**g**) Zn 2p signals of E-SiC-ZnFeZIF, and (**h**) Fe 2p signals of E-SiC-ZnFeZIF.

**Figure 10 nanomaterials-13-01664-f010:**
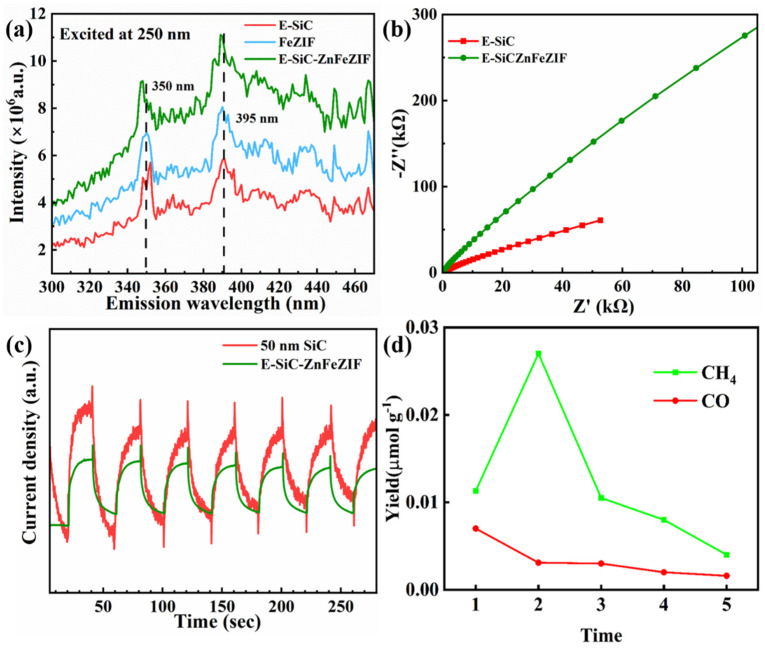
(**a**) PL spectroscopy, (**b**) electrochemical impedance spectroscopy, (**c**) photocurrent responses spectroscopy, (**d**) cycling yield for CH_4_ and CO production of E-SiC-ZnFeZIF. The CO_2_ reduction tests were repeated 5 times.

**Figure 11 nanomaterials-13-01664-f011:**
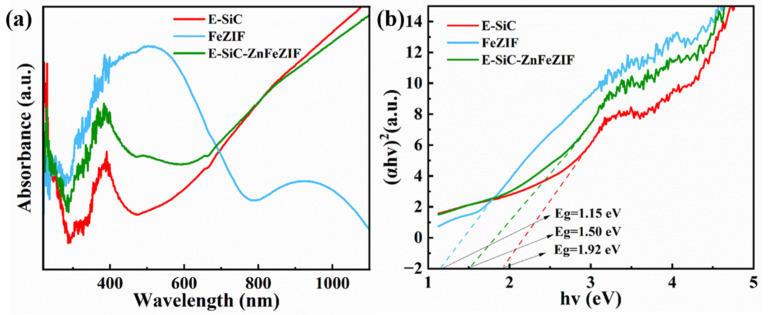
(**a**) UV–vis–NIR diffuse reflectance spectra of the prepared E-SiC, FeZIF, and E-SiC-ZnFeZIF; (**b**) the band gap energy of the prepared E-SiC, FeZIF, and E-SiC-ZnFeZIF.

**Figure 12 nanomaterials-13-01664-f012:**
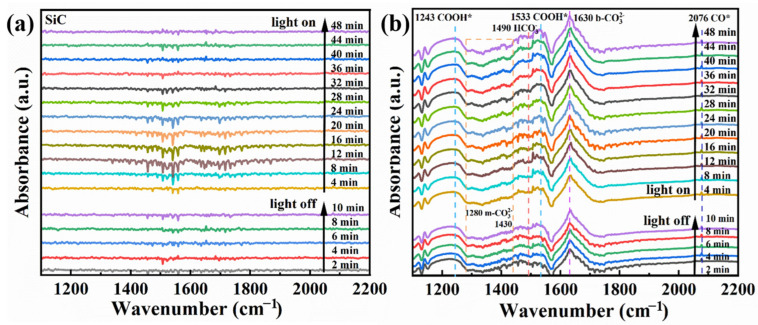
(**a**,**b**) In situ FTIR spectra of the E-SiC and E-SiC-ZnFeZIF after exposure to a mixture gas of CO_2_ and H_2_O in the dark for 10 min, and subsequent light illumination for 50 min.

**Figure 13 nanomaterials-13-01664-f013:**
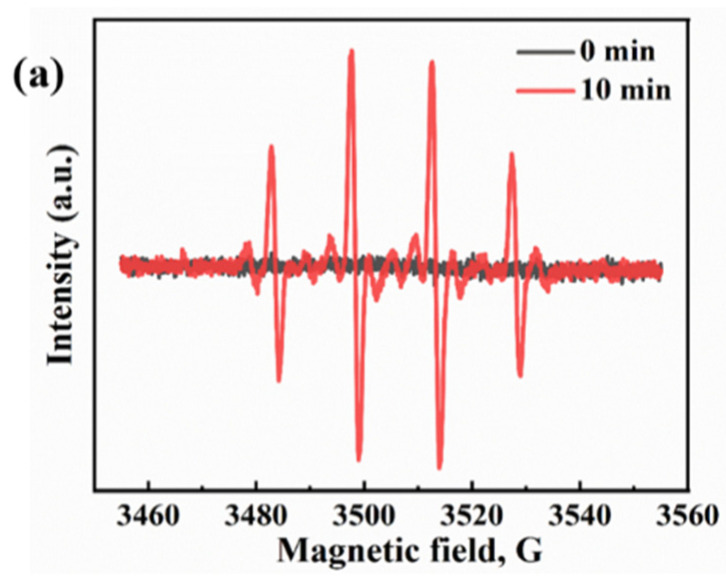
(**a**) •OH spectra of E-SiC-ZnFeZIF. (**b**) The mechanism of single metal reduction of carbon dioxide. (**c**) Scheme of the two possible paths for molecular CO_2_ and carbonate photoreduction. The symbol * indicates the species adsorbed on the photocatalyst surface. The same routes to the gas phase products (CH_4_, H_2_, and CO) also occur for the second path.

**Figure 14 nanomaterials-13-01664-f014:**
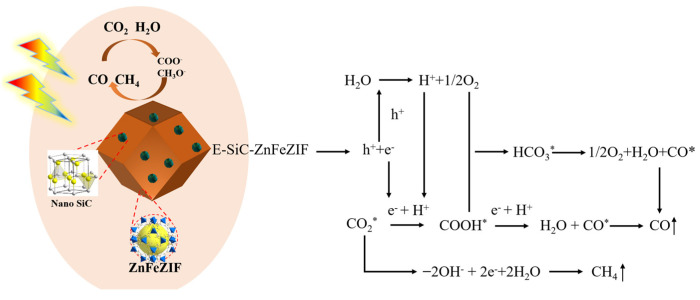
The proposed mechanism of photocatalytic reduction of CO_2_ into CO and CH_4_ by the E-SiC-ZnFeZIF.

## Data Availability

Not applicable.

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
