# Peer review of "SiC@FeZnZiF as a Bifunctional Catalyst with Catalytic Activating PMS and Photoreducing Carbon Dioxide"

_nanomaterials, 2023, doi:10.3390/nano13101664_

Round 1

Reviewer 1 Report (Previous Reviewer 2)

The authors have studied the SiC@FeZnZiF for the degradation of tetracycline (THC) and carbamazepine (CBZ) and the photocatalytic reduction of carbon dioxide.

My following points for the authors:

For both reactions is is important to clarify and perfomr the experiments for showing you are working in chemical kinetic regime, e.g. study the effect of stirring, amount of catalyst, particle size of the catalyst.

2. To study the meachnism you need to perform EPR studies and use of radical scavengers to support the mechanism you present. Figure 13a  has been added without detailed explanation and discussion in the text. I recommend the authors for EPR studies to read as a guidance the following paper:

https://pubs.acs.org/doi/full/10.1021/acscatal.6b00982

3. Use superscirpt in the document when you discuss and present radicals, the same for the figures. In some cases it is fine, in some other case it is not, so control the document.

5. For discussing the mechanism present time one line graphs showing conversion and yield of products.

6. Reusability of catalysts and leaching tests hsould be performed.

Overall, the manuscript has been improved but the authors have to perform significant amount of work.

Minor editing of language is needed.

Author Response

Reviewer 2 Report (New Reviewer)

Review of the manuscript SiC@FeZnZiF as a bifunctional catalyst with catalytic activating PMS and photoreducing carbon dioxide“  by  Zhiqi Zhua, Liaoliao Yanga, Zhaodong Xionga, DaoHan Liua, Binbin Hua, Nannan Wanga, Oluwafunmilola Olab and Yanqiu Zhuc  (Nanomaterials, MDPI)

This manuscript reports the experimental results related to the preparation, characterization, and testing of new photocatalysts based on SiC@FeZnZiF as potential catalysts for the photocatalytic degradation of tetracycline (THC) and carbamazepine (CBZ) as well as for the reduction of CO2. The catalysts were characterized by various instrumental techniques, such as SEM/EDS, XRD, FTIR, Raman analysis, XPS, UV-Vis –NIR diffuse reflectance spectroscopy and PL spectroscopy. The catalytic activities of the prepared photocatalysts were studied in a batch photoreactor under different reaction conditions and using 300 W xenon lamp as light source.

The work is quite extensive and contains many interesting results. This topic is of great interest because of its potential practical applications, and the authors have studied it in detail using various instrumental methods. In my opinion, the paper is technically correct and satisfactorily written in the first part, which deals with preparation and characterization of prepared photocatalysts. However, the second part of the paper, dealing with the interpretation of the results obtained for organic pollutants degradation and phptpcatalytic reduction of CO2 is not adequate. The obtained results are promising and innovative. However, the interpretation was not carried out in an appropriate manner. That is why I propose a detailed revision of the work.

 Specifically, I will try to make some comments to support this:

 1.  It is obvious that the photocatalytic measurements were performed in a batch photoreactor, but the experimental system should be described a little better. The schematic of the photocatalytic reaction device should be presented. Details related to the determination of reaction products are not given. More information should be provided about the measure of light intensity. How was the light intensity measured? Which were the obtained values? The distance between lamp and radiometer?

2.     What is the particle size of the photocatalyst prepared? It is recommended to add BET analysis and pore size distributions of photocatalysts.

3.     Chapter 2.1. (Chemicals and materials) should involve information about all the materials used in this work. The reagents purity should be stated. The base molecules of tetracycline (THC) and carbamazepine (CBZ) must be presented in the chemicals and materials section.

4.   There is inconsistency in the use of symbols and abbreviations and this should be corrected throughout the paper.

5.      Page 4, lines 148-150: „The 148 catalytic properties of the eSiC@FeZnZiF were estimated with 50 nm of commercial SiC 149 according to tetracycline hydrochloride and CBZ degradation.“ - This is not clearly written; please explain.

6.   Page 4, line 155: Is the pH adjusted and how? The zeta potential of the materials used must be determined. What are their isoelectric points IEP?

7.     Page 4, line 166: Why is sodium bicarbonate used; the specified compound is not on the list of chemicals and materials.

8.  Part of the work from Chapter 3.2. till the end of paper should be significantly improved. E.g. on Page 8: „In addition, the adsorption ability of ESiC-ZnFeZIF to CBZ and  THC molecules is poor.“ Where it can be seen? Speculation is not encouraged and all claims should be verified.; Page 9: „Therefore, 254 the high catalytic activity is mainly due to the activation of ZnFeZIF.“ Please, explain this in more details.; „The experimental results show that the ESiC-ZnFeZIF/PMS integrated system is fast and efficient in the mineralization of THC, confirming that ESiC-Zn-FeZIF has high catalytic degradation activity.“ The authors need to perform the experiments of total organic carbon removal to corroborate the degradation tests.

9.      Page 9, line 268: Instead Fig. 4 (e) it should be Fig. 4(d).

10.  Page 9: „When the concentration of PMS in the degradation environment was 2 mM, the 270 degradation efficiency slightly decreased compared to the aforementioned example.“ It is necessary to provide an explanation for such results.

11.  In the continuation of the work, there are still many speculations (it is hard to even list them) that need to be avoided, that is, the interpretation should be based on experimental results and evidence.

12.  It seems that the stability of the obtained photocatalysts is questionable. Explain how stability could possibly be improved.

13.  Authors should try to better describe the role of SiC. It is also necessary to more precisely and qualitatively describe the role of active species and radicals. Similar comments apply to the rest of the paper, but are not specified in more detail here.

14.  Page 17, line 497: reference numbers should be given.

15. The authors inconsistently use the abbreviations THC and TCH, etc. The manuscript contains many abbreviations. Please ensure that acronyms, abbreviations, and uncommon units are spelled out at the first mention in your manuscript file.
The results given in the tables in Sppl. material should be commented in the paper. Information given in Supp. materials are confusing.

16.  In conclussion, the authors have done a good job of describing the chemical aspects related to the preparation and characterization of advanced materials, but they have not done a good interpretation of the engineering aspects. Sometimes the authors describe the results without detailing the interpretation and the discussion.

17.  It is also necessary to carry out detailed proofreading of the text.

I believe that if all the above problems are addressed, more work is needed and the new version will eventually look quite different. Therefore, I propose a major revision of this article.

Review of the manuscript SiC@FeZnZiF as a bifunctional catalyst with catalytic activating PMS and photoreducing carbon dioxide“  by  Zhiqi Zhua, Liaoliao Yanga, Zhaodong Xionga, DaoHan Liua, Binbin Hua, Nannan Wanga, Oluwafunmilola Olab and Yanqiu Zhuc  (Nanomaterials, MDPI)

This manuscript reports the experimental results related to the preparation, characterization, and testing of new photocatalysts based on SiC@FeZnZiF as potential catalysts for the photocatalytic degradation of tetracycline (THC) and carbamazepine (CBZ) as well as for the reduction of CO2. The catalysts were characterized by various instrumental techniques, such as SEM/EDS, XRD, FTIR, Raman analysis, XPS, UV-Vis –NIR diffuse reflectance spectroscopy and PL spectroscopy. The catalytic activities of the prepared photocatalysts were studied in a batch photoreactor under different reaction conditions and using 300 W xenon lamp as light source.

The work is quite extensive and contains many interesting results. This topic is of great interest because of its potential practical applications, and the authors have studied it in detail using various instrumental methods. In my opinion, the paper is technically correct and satisfactorily written in the first part, which deals with preparation and characterization of prepared photocatalysts. However, the second part of the paper, dealing with the interpretation of the results obtained for organic pollutants degradation and phptpcatalytic reduction of CO2 is not adequate. The obtained results are promising and innovative. However, the interpretation was not carried out in an appropriate manner. That is why I propose a detailed revision of the work.

 Specifically, I will try to make some comments to support this:

 1.  It is obvious that the photocatalytic measurements were performed in a batch photoreactor, but the experimental system should be described a little better. The schematic of the photocatalytic reaction device should be presented. Details related to the determination of reaction products are not given. More information should be provided about the measure of light intensity. How was the light intensity measured? Which were the obtained values? The distance between lamp and radiometer?

2.    What is the particle size of the photocatalyst prepared? It is recommended to add BET analysis and pore size distributions of photocatalysts.

3.     Chapter 2.1. (Chemicals and materials) should involve information about all the materials used in this work. The reagents purity should be stated. The base molecules of tetracycline (THC) and carbamazepine (CBZ) must be presented in the chemicals and materials section.

4.   There is inconsistency in the use of symbols and abbreviations and this should be corrected throughout the paper.

5.      Page 4, lines 148-150: „The 148 catalytic properties of the eSiC@FeZnZiF were estimated with 50 nm of commercial SiC 149 according to tetracycline hydrochloride and CBZ degradation.“ - This is not clearly written; please explain.

6.   Page 4, line 155: Is the pH adjusted and how? The zeta potential of the materials used must be determined. What are their isoelectric points IEP?

7.     Page 4, line 166: Why is sodium bicarbonate used; the specified compound is not on the list of chemicals and materials.

8.  Part of the work from Chapter 3.2. till the end of paper should be significantly improved. E.g. on Page 8: „In addition, the adsorption ability of ESiC-ZnFeZIF to CBZ and  THC molecules is poor.“ Where it can be seen? Speculation is not encouraged and all claims should be verified.; Page 9: „Therefore, 254 the high catalytic activity is mainly due to the activation of ZnFeZIF.“ Please, explain this in more details.; „The experimental results show that the ESiC-ZnFeZIF/PMS integrated system is fast and efficient in the mineralization of THC, confirming that ESiC-Zn-FeZIF has high catalytic degradation activity.“ The authors need to perform the experiments of total organic carbon removal to corroborate the degradation tests.

9.      Page 9, line 268: Instead Fig. 4 (e) it should be Fig. 4(d).

10.  Page 9: „When the concentration of PMS in the degradation environment was 2 mM, the 270 degradation efficiency slightly decreased compared to the aforementioned example.“ It is necessary to provide an explanation for such results.

11.  In the continuation of the work, there are still many speculations (it is hard to even list them) that need to be avoided, that is, the interpretation should be based on experimental results and evidence.

12.  It seems that the stability of the obtained photocatalysts is questionable. Explain how stability could possibly be improved.

13.  Authors should try to better describe the role of SiC. It is also necessary to more precisely and qualitatively describe the role of active species and radicals. Similar comments apply to the rest of the paper, but are not specified in more detail here.

14.  Page 17, line 497: reference numbers should be given.

15. The authors inconsistently use the abbreviations THC and TCH, etc. The manuscript contains many abbreviations. Please ensure that acronyms, abbreviations, and uncommon units are spelled out at the first mention in your manuscript file. The results given in the tables in Suppl. material should be commented in the paper. Information given in Supp. materials are confusing.

16.  In conclussion, the authors have done a good job of describing the chemical aspects related to the preparation and characterization of advanced materials, but they have not done a good interpretation of the engineering aspects. Sometimes the authors describe the results without detailing the interpretation and the discussion.

17.  It is also necessary to carry out detailed proofreading of the text.

 In conclusion, I believe that if all the above problems are addressed, more work is needed and the new version will eventually look quite different. Therefore, I propose a major revision of this article.

Round 2

Reviewer 2 Report (New Reviewer)

Review of the manuscript SiC@FeZnZiF as a bifunctional catalyst with catalytic activating PMS and photoreducing carbon dioxide“  by  Zhiqi Zhua, Liaoliao Yanga, Zhaodong Xionga, DaoHan Liua, Binbin Hua, Nannan Wanga, Oluwafunmilola Olab and Yanqiu Zhuc  (Nanomaterials, MDPI)

The revised version of the manuscript is more acceptable than the original version. The authors successfully responded to the reviewers comments. Thus, I recommend acceptance of this manuscript for publication in the Nanomaterials. Congratulations on your work.

Best regards!

I have no serious objections on the Quality of English Language. Therefore, I suggest the editor to make a decision on whether the quality of the English Language needs improvement.

This manuscript is a resubmission of an earlier submission. The following is a list of the peer review reports and author responses from that submission.

Round 1

Reviewer 1 Report

MS No: 

nanomaterials-2322083

Title:

SiC@FeZnZiF as a bifunctional catalyst with combined advanced oxidation and photoreduction of carbon dioxide

Authors:     

Zhiqi Zhu, Liaoliao Yang, Zhaodong Xiong,Binbin Hu, Nannan Wang, Oluwafunmilola Ola and Yanqiu Zhu

The present manuscript deals with synthesis and characterization of SiC@FeZnZiF catalytic materials. Their activity was evaluated by degradation of tetracycline and  carbamazepine after PMS activation, as well as reduction of carbon dioxide. In my opinion, it can be accepted for publication in Nanomaterials after major revision.

·       The term “with combined advanced oxidation” in the title is not very specific. I would advise the authors to revise it.

·       The Authors should present data concerning metals leaching in the liquid phase.

·       Data concerning pharmaceuticals removal in the absence of catalyst as well as in the absence of PMS should be added in the revised manuscript.

·       Dara concerning the stability of the synthesized materials should be added in the revised manuscript. To do so, consecutive experimental runs should be carried out.

Reviewer 2 Report

The authors present the use of SiC@FeZnZiF for two reactions. One is the degradation of tetracycline (THC), carbamazepine (CBZ) and the second one is the photocatalytic reduction of carbon dioxide.

The paper needs significant improvement on the following points:

1. Detailed discussion of analytical protocols, calibrations and protocols for catalytic testing for both reactions. For example it is not clear the reaction setup for the two reactions, if the authors have studied important parameters such as effect of stirring and mass of catalyst to determine kinetic regime control and finally, which are the products for both reactions. For CO2 photocatalytic reduction the authorss should use also HPLC to determine liquid phase products. Moreover, they should present carbon balance, and formulas for calclulating activitye, conversion, yield and selectivity. Moreover, effect of pressure, amount of catalyst and temperature should be investigated.

2. Reusability of catalysts and characterisation of used catalysts.

3. Discussion about the active sites for both reactions and what is the connection for the structure-activity relationship for both reactions should be presented and discussed. The authors present data from characterisation they have done but there is no discussion. Band gap of the materials should be reported and compared the data with the literature.

4. Leaching exoeriments should be performed.

5. Some captions of the figures in the suppl. info are in chinese, not in Enlgish, the authors should check carefully the content and the manuscript should be checked carefully for grammatical errors and typos from a professional person who is expert in English.

I recommend major revision since the authors have to answer important questions and perform significant work for improving manuscript.

5. References should be added. There are important papers that have discussed CO2 photocatalytic reduction.

1. Catal. Sci. Technol., 2019, 9, 2253-2265

2. Appl. Catal., B, 2017, 200, 386–391 

Round 2

Reviewer 1 Report

Accept in present form

Reviewer 2 Report

Many of the comments need to be answered and the authors have to do the requested experiments to prove it.

1. Study of the kinetic regime has not been done.

2. Reusability and leaching tests should be for both reactions.

3. The authors have used only the literature to propose a reacton mechanism, they should perform experimental to work to justify the proposed mechanism.

4. The description of the figures in the suppl. info needs to be improved.

5. The use of Enlgish is poor, the document needs to be checked thorougly from an expert, ask MDPI.
